# Co-Occurring X-Linked Agammaglobulinemia and X-Linked Chronic Granulomatous Disease: Two Isolated Pathogenic Variants in One Patient

**DOI:** 10.3390/biomedicines11030959

**Published:** 2023-03-21

**Authors:** Lauren Gunderman, Jeffrey Brown, Sonali Chaudhury, Maurice O’Gorman, Ramsay Fuleihan, Aaruni Khanolkar, Aisha Ahmed

**Affiliations:** 1Division of Allergy and Immunology, Ann & Robert H. Lurie Children’s Hospital of Chicago, Chicago, IL 60611, USA; 2Department of Pediatrics, Feinberg School of Medicine, Northwestern University, Chicago, IL 60611, USA; 3Division of Gastroenterology, Ann & Robert H. Lurie Children’s Hospital of Chicago, Chicago, IL 60611, USA; 4Division of Hematology, Oncology and Stem Cell Transplantation, Ann & Robert H. Lurie Children’s Hospital of Chicago, Chicago, IL 60611, USA; 5Department of Pediatrics and Pathology, Keck School of Medicine, University of Southern California, Los Angeles, CA 90027, USA; 6Division of Pathology and Laboratory Medicine, Children’s Hospital Los Angeles, Los Angeles, CA 90027, USA; 7Division of Allergy, Immunology and Rheumatology, Department of Pediatrics, Columbia University Irving Medical Center, New York Presbyterian Morgan Stanley Children’s Hospital of New York, New York, NY 10032, USA; 8Department of Pathology, Ann & Robert H. Lurie Children’s Hospital of Chicago, Chicago, IL 60611, USA; 9Department of Pathology, Feinberg School of Medicine, Northwestern University, Chicago, IL 60611, USA

**Keywords:** XLA, BTK, X-linked chronic granulomatous disease, X-linked agammaglobulinemia, HSCT, multi-genetic diagnosis

## Abstract

We present a unique and unusual case of a male patient diagnosed with two coexisting and typically unassociated X-linked conditions: he was initially diagnosed with X-linked agammaglobulinemia (XLA) followed by a diagnosis of X-linked chronic granulomatous disease (XCGD) and an as of yet unpublished hypomorphic gp91phox variant in the *CYBB* gene. The latter was tested after the finding of granulomatous gingivitis. Hematopoietic stem cell transplant (HSCT) was performed due to severe colitis and nodular regenerative hyperplasia (NRH) of the liver. Following transplant, complete donor engraftment was observed with the restoration of a normal oxidative burst and full restoration of normal levels of circulating, mature CD19+ B cells. This case is singular in that it does not involve a contiguous gene syndrome in which deleted genes are in close proximity to either *BTK* and *CYBB,* which has been previously reported. To our knowledge, this is the first reported case of XLA and XCGD co-existing in a single patient and of having both inborn errors of immunity successfully treated by HSCT.

## 1. Introduction 

X-linked agammaglobulinemia (XLA) and X-linked chronic granulomatous disease (XCGD) are two of the five most common X-linked immunodeficiencies, both often diagnosed via single- or multi-gene Sanger sequencing despite the rapidly increasing use of next-generation sequencing [1]. Disregarding a shared location on the X chromosome, the diseases themselves are distinct: XLA is an antibody deficiency syndrome caused by mutations in the Bruton tyrosine kinase (*BTK*) gene leading to deficiency of mature B lymphocytes, agammaglobulinemia, and susceptibility to infection [1,2]. Immunoglobulin replacement continues to be the cornerstone of treatment, with improved outcomes in cases of early diagnosis and treatment initiation [3]. Alternatively, the X-linked form of CGD is due to mutations in the *CYBB* gene which encodes the gp91-phox component of the NADPH oxidase complex. *CYBB* is affected in 65–75% of total CGD cases in western countries and is one of six known genes coding building blocks of the NADPH oxidase complex [4]. In XCGD, the resulting issue with respiratory burst is more severe than in autosomal recessive disease states, leading to increased infections [4]. Survival and patient outcomes have improved with the introduction of antimicrobial prophylaxis and, in some countries, anti-interferon therapy [5]. Yet, to achieve the cure, HSCT is necessary and is often pursued for control of non-infectious autoinflammatory complications [4,5,6].

## 2. Case Presentation

A Caucasian boy presented at 24 months of age with a history of recurrent sinopulmonary infections and croup. He was fully vaccinated and otherwise healthy, without hospital admissions, poor growth, or developmental delay. Parents and younger sisters were healthy, with no known history of diseases in immediate or extended family members. On laboratory evaluation, he had undetectable serum IgG, IgA, and IgE; low IgM, [28.9 mg/dL, reference range (RR): 46–160 mg/dL] and <1% circulating CD19+ B cells [<56 cells/mm^3^ absolute CD19+ B cells; RR for children from 691–1115 days of life: 17–34% of lymphocytes and 523–1779 cells/mm^3^] by flow cytometry, raising suspicion for XLA or an alternative early defect in the development of B-cells. Quantitation of CD3+, CD4+, CD8+ T cells, and natural killer (NK) cells was within normal limits. He started intravenous immunoglobulin replacement (IVIG) at 0.5 g/kg (7.5 g) every 4 weeks with adjustments to achieve infection control.

In considering the differential diagnosis of this patient, the importance of understanding B-cell development cannot be understated. B-cell development from lymphoid progenitors is orchestrated via the expression of various transcription factors such as Pax5 (a B-cell commitment factor), TCF3, and Ikaros. Additionally, the development of the B cell receptor (BCR) involves several stages, including the formation of the pre-BCR, which is crucial for the first checkpoint in the pre-BII stage. Mutations in these transcription factors or mutations in genes affecting pre-BCR signaling (such as BTK, IGM, CD79a/b, BLNK, L14.1, or PI3Kα) can lead to early B cell defects and disorders such as XLA or autosomal recessive agammaglobulinemia [7,8]. As the patient was a boy with agammaglobulinemia, a targeted genetic evaluation with the sequencing of the Bruton’s tyrosine kinase (*BTK*) gene was performed. Genomic DNA from the X-chromosome was analyzed by PCR/DHPLC (denaturing high-performance liquid chromatography) for mutations in *BTK*. Analysis of exon 3 demonstrated the insertion of a single nucleotide (A) in a string of 7 A’s in codons 70–72. This frameshift mutation clearly confirmed the diagnosis of XLA. The patient’s biological mother was determined to be a carrier of the variant. Following sequencing, flow cytometry was performed using an in-house clinically validated BTK assay, which revealed a lack of BTK expression in monocytes (Figure 1), further supporting the diagnosis of XLA.

Within the first three years following diagnosis, the patient’s infection frequency improved on IVIG despite intermittent breakthrough infections with sinusitis in the winter months and a single episode of otitis media. During these years, he also developed stomatitis inside and around the mouth, with swollen lips and gums. Thrush was noted at the time (despite normal to elevated levels of CD3+, CD4+, and CD8+ T cells and NK cells) and responded to topical treatment with Nystatin. With the stomatitis, neutrophil dysfunction was considered given the known association with neutropenia in untreated XLA; but the family was consistent with his replacement immunoglobulin, and the neutrophil count remained in the normal range [9,10]. Due to the persistence of the aphthous ulcers, he was treated empirically with Acyclovir due to concern that these may be herpetic lesions. The lesions persisted for over 4 weeks, only improving after increased immunoglobulin levels (goal of >1000 mg/dL) and as-needed topical Acyclovir cream. At age 5, he developed non-tender submandibular lymphadenopathy. Laboratory assessment was reassuring, with an in-range complete blood count, differential leucocyte count, lymphocyte subsets, and undetectable EBV and CMV viremia by PCR testing. Antibody testing was avoided due to routine immunoglobulin replacement and the diagnosis of antibody deficiency in the setting of XLA. Ultrasound of the neck showed multiple prominent bilateral lymph nodes without a clear etiology. No biopsies were taken, but continued monitoring was planned. At 6 years of age, he developed painful gingivitis. The submandibular lymphadenopathy had persisted and was now thought to be reactive. The inflamed gingival tissue was biopsied and cauterized by the patient’s oral surgeon. Histopathological analyses of the tissue revealed “mucosal ulceration with focal granulomatous inflammation, hyperparakeratosis and acanthosis”, consistent with granulomatosis gingivitis (Figure 2). Once the inflamed tissue was removed, his lymphadenopathy resolved. Due to the pathology revealing granulomas, functional testing by flow cytometry [11] was completed for chronic granulomatous disease and showed a reduced but not absent neutrophil oxidative burst (DHR assay) (Figure 3). This result raised the question of a false positive result due to the *BTK* variant, as Btk is a known negative regulator of the neutrophilic oxidative burst [12]. The resulting high levels of NADPH activity from pathogen stimulation in Btk-deficient neutrophils have been shown to cause neutrophil apoptosis in patients with XLA [12]. On review, this patient’s neutrophils were in the normal range. The patient started daily Trimethoprim/Sulfamethoxazole prophylaxis while maternal testing was completed to determine the inheritance pattern prior to genetic testing. His mother’s oxidative burst assay revealed a bimodal DHR fluorescence pattern due to two populations of neutrophils, one with slightly decreased DHR fluorescence (DHR+) and one with normal DHR fluorescence (DHR++) (Figure 3), indicating a hypomorphic variant XCGD maternal carrier state. Targeted sequencing of *CYBB* was completed in the patient, which revealed an unpublished likely pathogenic hemizygous missense variant [c.272G > T (p.Arg91Leu)], confirming the diagnosis of XCGD [13]. The variant was assumed to be hypomorphic given the markedly decreased but not absent oxidative burst. An increase in infections and autoimmunity may be present in X-linked CGD carriers, most strongly associated with carriers that have less than 20% oxidative burst; therefore, maternal history was assessed [14]. Fortunately, the patient’s mother was asymptomatic, which is consistent with random inactivation of the X-chromosome as demonstrated in her DHR assay (Figure 3).

Apart from occasional localized gingival swelling, after a year on antibiotic prophylaxis and continued immunoglobulin replacement, he was doing well with decreased infection frequency. At age 8, he developed constipation, stomach aches, and blood in the stool, which initially contributed to an anal fissure. There was minimal improvement with polyethylene glycol and mesalamine. He had rectal bleeding, poor weight gain, and elevated fecal calprotectin (>2000 mcg/g). Esophagogastroduodenoscopy (EGD) was overall reassuring, but colonoscopy showed a rectal stricture and narrowing. Biopsies revealed active chronic inflammation, crypt distortion, abscess formation, ulceration, and granulomas (Figure 2), consistent with Crohn’s-like colitis, particularly in the setting of intense, focal areas of disease in the descending and sigmoid colon and rectum. He began treatment simultaneously for Giardia infection and colitis with steroid enemas, metronidazole, intermittent mesalamine, and eventually 6-mercaptopurine (6-MP). He had continued weight loss and decreased height velocity and was unable to tolerate methotrexate. As a result, biologic medications were pursued, with improvement of the colitis on vedolizumab (anti-α4β7 monoclonal antibody). However, he continued to have progressive strictures and an on-going growth and development delay.

In addition to colitis, he was monitored closely for persistent transaminitis. An abdominal CT showed liver enlargement. Liver biopsy was positive for nodular regenerative hyperplasia (NRH) with changes suggestive of hepatoportal sclerosis with negative staining for pathogens. A few small, non-necrotizing granulomas were also present (Figure 2). While most often occurring in common variable immunodeficiency, NRH is a reported complication of CGD and a frequent, yet underreported, late-onset complication of XLA (median age of presentation is 20 years), which can lead to non-cirrhotic portal hypertension and is associated with increased morbidity and mortality. On review of 21 patients with XLA, Nunes-Santos et al. describe elevated alkaline phosphatase, decreased platelet counts (<100,000 per uL for >6 months), and hepatosplenomegaly as indicators for +NRH biopsies. All findings, apart from low platelet counts, were consistent in this patient [15,16]. Although asymptomatic, given worrisome liver findings, on-going transaminitis, and a history of severe Crohn’s-like colitis with continued growth failure, it was recommended that the patient proceeds to HSCT.

At 16 years of age, HSCT was performed with an 8/10 HLA mismatched-DR, DQ LRMM unrelated donor following reduced intensity conditioning (Busulfan/Fludarabine/ATG) utilizing peripheral blood stem cells. Complications included EBV viremia, treated preemptively with Rituximab, and Coombs positive warm autoimmune hemolytic anemia, treated with steroids followed by a successful wean. Post-transplant laboratory findings showed complete normalization of the fecal calprotectin from 3020 mcg/gr pre-SCT to 36 mcg/gr post-SCT (which has sustained −34 mcg/gr 9 months after HSCT), normalized neutrophilic oxidative burst (Figure 2), and normal numbers of circulating CD19+ B cells. However, due to prolonged hypogammaglobulinemia post-transplant, likely secondary to Rituximab infusions, regular immunoglobulin replacement was restarted. The patient is currently doing well on monthly IVIG with no significant infections and no return of GI symptoms or complaints. Liver biopsy has not been repeated.

## 3. Discussion

We present a unique case of co-existing variants, one pathogenic and the other hypomorphic, in a single patient, leading to two separate X-linked conditions, XLA and XCGD. Unlike in contiguous gene deletion syndromes, defined as the deletion of a segment that contains two or more adjacent genes, these variants are isolated and singular. The X-chromosome, which is approximately three times larger than the Y chromosome, contains 900 genes, including *CYBB* (located on Xp21.1) and *BTK* (mapped to the Xq21.3-Xq22 region) [17]. Both *CYBB* and *BTK* have been associated with large deletions leading to co-existing genetic diagnoses with other genes: Lhomme et al. describe this rare association of large deletions containing additional genes in eight XCGD patients with McLeod phenotype (neuroacanthosis) and Duchene muscular dystrophy (DMD) or Retinitis Pigmentosa (RP) [18,19], and in XLA, large deletions have included a neighboring gene to *BTK* (*TIMM8A*) causing Mohr–Tranebjaerg syndrome, an X-linked sensorineural hearing loss impairment in childhood with progression to neurodegeneration [20]. Such examples are important to identify in order to understand differences in the genetics of our patient from those previously reported, as the *CYBB* and *BTK* variants in the patient we described are neither large deletions nor impacting each other, as in the case of neighboring genes.

As inflammatory complications occur in XLA despite treatment with immunoglobulin replacement, the mouth sores and lymphadenopathy in this case may not have been outside normal expectations for XLA. Patients with hypogammaglobulinemia can also continue to have breakthrough sinus infections, again which may be from XLA alone or from the additional defect in the *CYBB* gene. This additional defect hypothesis is further supported by the discovery of the granulomatous gingival lesions, which raised concern for CGD and led to additional testing when the patient was 5 years old. Reduced but not absent, the neutrophilic oxidative burst in this patient with a known *BTK* variant was initially questioned. Both the possibility of a hypomorphic variant and the effect of Btk on NADPH activity were discussed. Using samples from patients with XLA, Honda et al. describe Btk’s role as a negative regulator of the oxidative burst, demonstrating that the absence of Btk leads to increased reactive oxygen species (ROS) formation from increased NADPH activity. When pathogens stimulate NADPH activity, the high levels of ROS lead to increased neutrophil apoptosis. Thus, one could hypothesize that a pathogenic variant in Btk leading to Btk-deficient neutrophils could increase the production of ROS, lead to apoptosis of Btk-deficient neutrophils, and, as a result, reduce neutrophilic oxidative burst. Of note, this was the opposite of what the group described in mice where ROS failed induction [12]. Interestingly and similarly, the use of BTK inhibitors, a treatment for B-cell malignancies and autoimmunity, has resulted in neutropenia and increased infections in some cases [21]. These examples show us there is still much to learn about Btk’s regulation of the oxidative burst and the causative effect of its absence on the apoptosis of neutrophils. In our patient, the diagnosis of XCGD was further supported and confirmed by the presence of granulomatous findings, maternal testing showing two populations of neutrophils with oxidative burst activity, and genetic testing showing a likely pathogenic variant in the *CYBB* gene.

Following confirmation of a second genetic diagnosis (XCGD) in the patient, he developed gastrointestinal symptoms concerning Crohn’s disease or CGD colitis. Assuming at the time that the GI issues were secondary to XCGD due to the time of onset and granulomatous colitis, it is possible that the patient’s XLA was contributing to his enteropathy. XLA, in the past, thought easily managed and complications avoided with rapid initiation of immunoglobulin replacement, now has a known association with autoimmunity/inflammatory conditions, including diarrhea and IBD-like enteropathy [22]. Hence, XLA could be an alternative etiology of the patient’s GI disease, or when considering the patient has two disease-causing variants, perhaps the severity of the patient’s intestinal inflammation was due to a digenic effect of both affected genes. This idea of a mixed phenotype is further supported in Stray-Pedersen et al., where a large effort was undertaken to examine the ability of whole exome screening methods in detecting disease-causing variants in patients with primary immunodeficiency diseases [23]. During this process, out of 278 families from 22 countries, 5% of patients were discovered to have two disease-causing genes/variants, leading to a mixed phenotype [23]. This broadened the idea that a “mutational burden” could alter a patient’s phenotype, lead to changes in management, and, as in our patient, contribute to a complex phenotype [23]. At the time of the patient’s diagnosis, extensive genetic panels were not available. This case demonstrates the value of performing primary immunodeficiency gene panels for the diagnosis of inborn errors of immunity.

In addition to genetic sequencing, functional laboratory evaluation was imperative in the patient’s diagnosis of XLA and XCGD. The initial diagnosis of XLA was suspected due to recurrent infections, agammaglobulinemia, and absent CD19 B cells. Single-gene sequencing and BTK expression by flow cytometry confirmed the diagnosis, with flow cytometry showing the absence of BTK expression in monocytes (Figure 1). Granulomas are a common complication of CGD and can be part of the initial presentation. Although we cannot be sure that the patient’s CGD was not involved in his initial presentation at age 24 months with recurrent sinopulmonary infections, we propose the age of onset of his CGD to be either 24 months or between 4 and 5 years of age when the stomatitis developed. Based on the clues from granulomatous lesions and maternal neutrophilic oxidative burst testing (Figure 3), targeted genetic sequencing for the *CYBB* gene was completed. From a combination of maternal functional and genetic testing, we can hypothesize that the variants were inherited from a single maternal X-chromosome and are not likely to be secondary to de novo variant occurrence.

Patients with CGD (X-linked and autosomal recessive) have been cured of their disease with HSCT. The same can be said for XLA in certain countries outside the United States where curative HSCT is the more manageable, cost-effective option when compared to life-long, regular immunoglobulin infusions. Outcomes for our patient have overall been positive: his XCGD is cured, as demonstrated by a normalized post-transplant oxidative burst (Figure 2), and he has resolution of the agammaglobulinemia. For now, he has required continued immunoglobulin replacement, suspected to be from the necessary use of anti-CD20 therapy (Rituximab) for post-transplant EBV viremia. The hope is that the need for immunoglobulin replacement may be temporary.

This case brings up thought-provoking questions about a possible mechanism for co-occurring XLA and X-CGD, which requires further study, and the development of new treatment strategies for patients with inborn errors of immunity (IEI). Among the emerging treatment options is gene therapy, a cutting-edge next step in curing our patients. However, in current studies, the goal of single-gene targeted treatment would not cure our patient or similar patients with more than one pathogenic defect. This case supports the continued importance of considering HSCT in treating severe, multi-gene cases of IEI, although one should also bear in mind the prevailing evidence indicating a high mortality rate associated with HSCT in patients with certain types of humoral immune defects [24]. Furthermore, as part of the genetic counseling, we must also begin to consider discussing the possibility of additional variants being present in patients evaluated by targeted or paneled sequencing, as this may impact treatment options.

## 4. Conclusions

Despite a previously proven genetic diagnosis, immunologists should remain alert for symptoms that could indicate additional genetic lesions. This is especially important as management may be altered based on the additional information. In this patient case, in the United States and at our center, XLA is typically managed with immunoglobulin replacement and close monitoring for infectious or auto-immune complications. However, HSCT is curative for XCGD and can reverse inflammatory complications such as inflammatory bowel disease. The case teaches the following lessons: the importance of continued surveillance for immune dysregulation issues post-diagnosis even in patients on indicated therapy, the value of functional testing in addition to genetic sequencing when the diagnosis is not clear, and the vitality of interdisciplinary collaboration in the medical (including dental) and laboratory spaces in forming diagnoses and optimizing patient care.

## Figures and Tables

**Figure 1 biomedicines-11-00959-f001:**
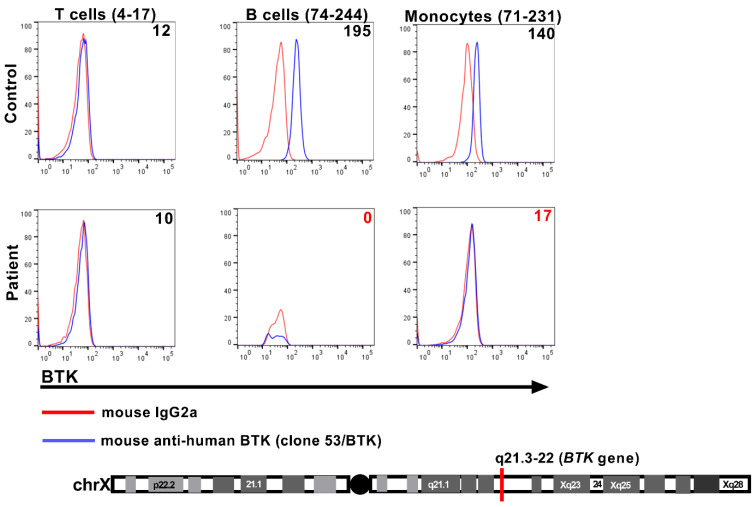
Assessment of cytosolic BTK expression by flow cytometry. The histogram overlay plots display the presence and/or absence of BTK protein expression (blue histograms) in the cytosol of the indicated cellular subsets. The numbers listed in parentheses adjacent to the cell subset label reflect the clinically validated reference ranges (5–95th percentiles) for that particular cell subset, derived from the background-subtracted median fluorescence intensity (MFI) of BTK expression evaluated for 34 healthy control donors. The numbers listed within each histogram overlay plot reflect the background-subtracted MFI of BTK expression for that specific cell subset for each donor. The specificity of the BTK signal was determined by also separately staining the cells with a dose-matched, isotype control Ab (red histograms). T cells lack BTK expression and serve as an internal specificity control in the assay.

**Figure 2 biomedicines-11-00959-f002:**
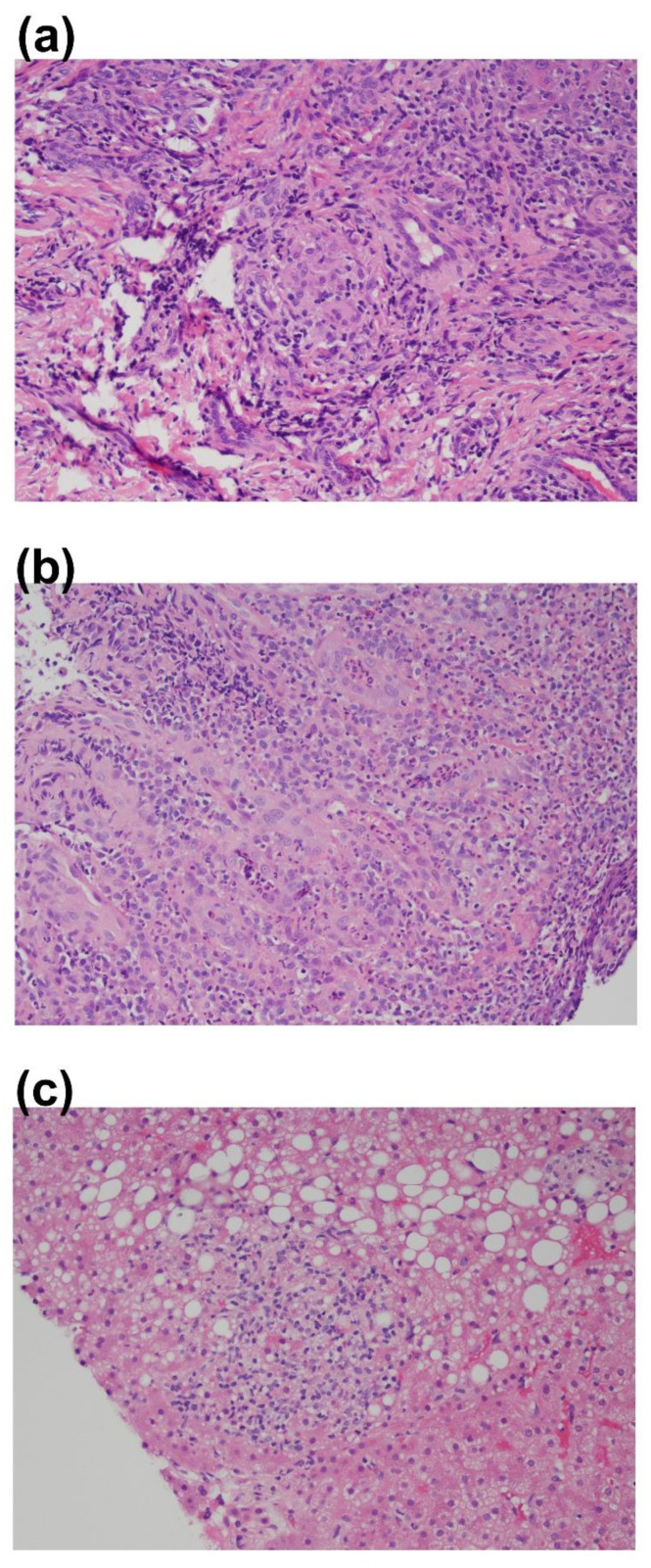
Histopathological evidence of granulomatous inflammation. (**a**) Mucosal ulceration with focal granulomatous inflammation, hyperparakeratosis, and acanthosis: image is focused on gingival tissue at the large zone of ulceration. Within fibrous connective tissue, a heavy mixed inflammatory cellular infiltrate that is predominantly neutrophils, lymphocytes, and histiocytes is seen. Widely scattered ill-defined clusters of histiocytes and multinucleated giant cells are noted. (**b**) Biopsy of transverse colon tissue showing large bowel mucosa with few granulomas. No significant crypt distortion. (**c**) Biopsy of the liver performed via needle biopsy. Hepatic architecture shows subtle distortion with nodular areas of widened alternating with areas of narrowed liver cell plates. No significant portal inflammation or interface hepatitis. Many of the portal areas lack a distinctive portal vein, and there are occasional foci of lobular inflammation. There is mild focal steatosis, involving less than 5% of parenchyma.

**Figure 3 biomedicines-11-00959-f003:**
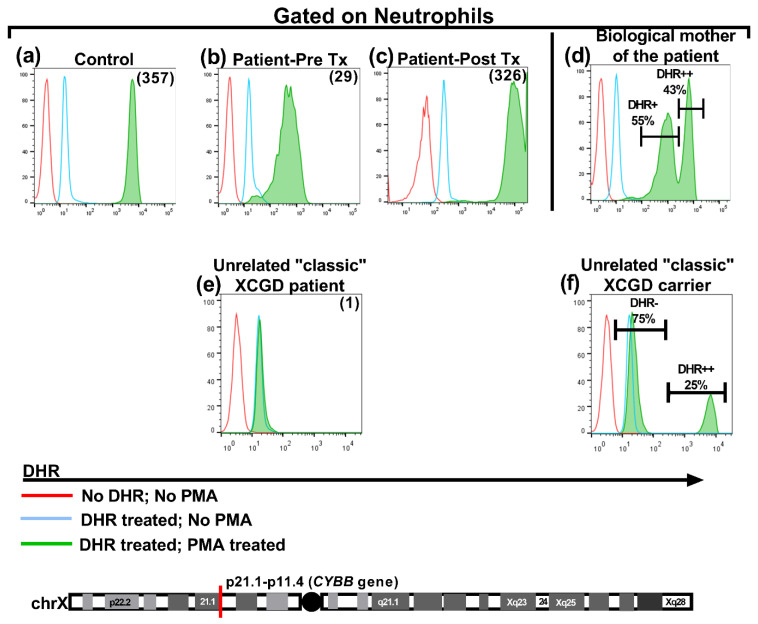
Assessment of neutrophil oxidative burst activity by flow cytometry. Three aliquots of whole blood from each donor were utilized to evaluate neutrophil oxidative burst activity. The first aliquot was left untreated. The second aliquot was labeled with dihydrorhodamine (DHR) and left unstimulated. The third aliquot was labeled with DHR and stimulated with phorbol 12-myristate 13-acetate (PMA) to induce neutrophil oxidative burst activity. The numerical values listed in parentheses in the overlaid histogram plots depict the neutrophil oxidative index (NOI), which is derived by dividing the median fluorescence intensity (MFI) of Rhodamine 123 observed in the PMA-stimulated tube (third aliquot) by the MFI of DHR measured in the second aliquot (unstimulated tube) (normal NOI >30). The DHR fluorescence patterns for the healthy control (**a**) and the patient sample pre- and post-stem cell transplant (**b**,**c**) are displayed. The DHR fluorescence in the biological mother’s sample (**d**) displays a “variant” bimodal distribution XCGD carrier pattern on account of the hypomorphic *CYBB* variant and random inactivation of the X-chromosome (lyonization). The frequencies of the maternal neutrophils harboring the normal (wild-type) and variant (mutated) *CYBB* genes are listed adjacent to the appropriate histogram peaks (DHR++ and DHR+). As a point of reference, the DHR fluorescence patterns from an unrelated case of “classic” XCGD (**e**) and a “classic” XCGD carrier (**f**) are also displayed. The frequencies of the neutrophils harboring the normal (wild-type) and mutated *CYBB* gene for the classic XCGD carrier are listed adjacent to the appropriate histogram peaks (DHR++ and DHR-) Tx: transplant.

## Data Availability

All relevant data was shared in this manuscript.

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
