# Peer review of "Co-Occurring X-Linked Agammaglobulinemia and X-Linked Chronic Granulomatous Disease: Two Isolated Pathogenic Variants in One Patient"

_biomedicines, 2023, doi:10.3390/biomedicines11030959_

Round 1
Reviewer 1 Report
Thank you for submitting your valuable case report. This paper is well written and worthy of publication. I have 3 comments.
1. Please describe the child's immunization history.
2. Please add the history of infection in the child after starting IVIG to XLA.
3. At what age do the authors consider the onset of CGD in the child?
Reviewer 2 Report
The case of the male patient with coexisting XLA and XCGD is a valuable and rare case for further studies that are clinically appealing. The successful treatment of both conditions with HSCT suggests that this could be a potential treatment option for other patients with similar conditions. This case can also serve as a starting point for future research into the genetic and immunological mechanisms underlying the co-occurrence of XLA and XCGD, which may help to identify potential therapeutic targets for these conditions.
Furthermore, the identification of the hypomorphic gp91phox variant in the CYBB gene in this patient highlights the importance of genetic testing in patients with complex immunological disorders. Understanding the genetic basis of these conditions can help with early diagnosis, personalized treatment, and genetic counseling for affected individuals and their families. This case also emphasizes the importance of considering the possibility of coexisting genetic conditions in patients with complex clinical presentations, even if they are not typically associated.
In summary, this case report provides important insights into the treatment and genetic basis of XLA and XCGD and highlights the need for further research in this area. The findings from this case could have implications for the development of new treatment strategies and genetic counseling for patients with similar conditions. Please comment.
This reviewer personally misses some insights regarding the following points:
The development of B-cells from lymphoid progenitors involves the expression of various transcription factors, including PAX5, which is a B-cell commitment factor. Mutations in early transcription factors such as TCF3, Ikaros, and Pax5 can lead to early B-cell deficiencies. The development of the BCR involves several stages, including the formation of the pre-BCR, which is crucial for the first checkpoint in the Pre-BII stage. Mutations in genes such as BTK, IGM, CD79a/b, BLNK, L14.1, or PI3Kalpha can affect pre-BCR signaling and lead to disorders such as XLA or AR. In contrast, mutations in genes such as TCF3, Ikaros, and Pax5 lead to early B-cell defects. The case of the male patient underscores the importance of hematopoietic stem cell transplantation (HSCT) in treating severe cases of inborn errors of immunity. Future studies can use these insights to develop new treatment strategies for patients with inborn errors of B-cell immunity, please expand referring to 10.1016/B978-0-12-818731-9.00124-5
Round 2
Reviewer 2 Report
The authors have clarified several of the questions I raised in my previous review. Most of the major problems have been addressed by this revision.